# Glucose and Oxygen Levels Modulate the Pore-Forming Effects of Cholesterol-Dependent Cytolysin Pneumolysin from *Streptococcus pneumoniae*

**DOI:** 10.3390/toxins16060232

**Published:** 2024-05-21

**Authors:** Michelle Salomé Hoffet, Nikola S. Tomov, Sabrina Hupp, Timothy J. Mitchell, Asparouh I. Iliev

**Affiliations:** 1Institute of Anatomy, University of Bern, Baltzerstrasse 2, 3012 Bern, Switzerland; michelle.hoffet@students.unibe.ch (M.S.H.); nikola.tomov@unibe.ch (N.S.T.); sabrina.hupp@unibe.ch (S.H.); 2School of Immunity and Infection, College of Medical and Dental Sciences, University of Birmingham, Edgbaston, Birmingham B15 2TT, UK; t.j.mitchell@bham.ac.uk

**Keywords:** ischemia, hypoxia, hypoglycemia, *Streptococcus pneumoniae*, pneumolysin, brain, astrocytes, microglia, pore formation, transient pores

## Abstract

A major *Streptococcus pneumoniae* pathogenic factor is the cholesterol-dependent cytolysin pneumolysin, binding membrane cholesterol and producing permanent lytic or transient pores. During brain infections, vascular damage with variable ischemia occurs. The role of ischemia on pneumolysin’s pore-forming capacity remains unknown. In acute brain slice cultures and primary cultured glia, we studied acute toxin lysis (via propidium iodide staining and LDH release) and transient pore formation (by analyzing increases in the intracellular calcium). We analyzed normal peripheral tissue glucose conditions (80 mg%), normal brain glucose levels (20 mg%), and brain hypoglycemic conditions (3 mg%), in combinations either with normoxia (8% oxygen) or hypoxia (2% oxygen). At 80 mg% glucose, hypoxia enhanced cytolysis via pneumolysin. At 20 mg% glucose, hypoxia did not affect cell lysis, but impaired calcium restoration after non-lytic pore formation. Only at 3 mg% glucose, during normoxia, did pneumolysin produce stronger lysis. In hypoglycemic (3 mg% glucose) conditions, pneumolysin caused a milder calcium increase, but restoration was missing. Microglia bound more pneumolysin than astrocytes and demonstrated generally stronger calcium elevation. Thus, our work demonstrated that the toxin pore-forming capacity in cells continuously diminishes when oxygen is reduced, overlapping with a continuously reduced ability of cells to maintain homeostasis of the calcium influx once oxygen and glucose are reduced.

## 1. Introduction

Bacterial infections by *S. pneumoniae* carry major public health significance, with meningitis being the most serious form accompanied with a high lethality in many patients and long-term disabilities in survivors [1]. While in Western countries meningitis prevalence is relatively low, in some regions of the world, its incidence is high and the social burden is substantial [2].

Multiple factors play a role in the pathogenesis of pneumococcal meningitis and pneumococcal infections, as the critical neuropathological factor is the cholesterol-dependent cytolysin pneumolysin (PLY) [3]. Pneumococci without PLY cause milder diseases—meningitis, sepsis, or pneumonia [3,4,5]. In meningitis, for example, the level of PLY in the cerebrospinal fluid (CSF) correlates well with the disease’s severity and prognosis [6]. The toxin is a 65 kD protein, binding membrane cholesterol and forming pre-pores and pores of 30–50 monomers in cell membranes [7]. In high concentrations, PLY induces host cell lysis [8]. At lower concentrations, it does not kill cells, which repair their membrane damage, but causes structural and functional changes through cytoskeletal reorganization, altered cell trafficking, and proinflammatory effects [8,9,10]. One of the major pathogenic mechanisms involved in PLY-mediated toxicity is the calcium influx through non-lytic membrane pores, contributing to delayed secondary cell damage [11].

Tissue damage in meningitis reveals various features, such as hippocampal apoptosis [11], cortical synapse loss [12], and axonal damage [13]. Computer tomography analysis demonstrates cerebrovascular complications in 37% of all patients [14], while histopathological evidence indicates even a higher incidence of brain ischemia [15]. Vascular obstruction can lead to complete oxygen and glucose tissue deprivation, producing infarction (observed variably in pneumococcal meningitis [15,16]) close to the obstructed vessels, or partial brain ischemia (as judged by the HIF-1α expression without infarction) further outside the immediate areas of vessel damage. Vascular damage is observed in non-neural tissues too [17]. As a result of ischemic infarction, a central tissue core with irreversible necrotic damage around the obstructed vessel is observed. The surrounding tissue outside the necrotic core is called penumbra, where the tissue is exposed to partial hypoxia and glucose deprivation but is still not irreversibly damaged and can recover upon re-established blood supply [18]. In meningitis, despite successful antimicrobial and antiedematous therapies, brain ischemia remains largely underestimated and is still not effectively addressed [19].

In this work, we address the crosstalk between PLY lytic and non-lytic pore-forming effects and their modulation via ischemic changes in the tissue, modeling closely the real tissue parameters of ischemia inside and outside the brain, namely hypoxia and hypoglycemia.

## 2. Results

In dissociated primary glial cultures and acute brain slices, we modeled the components (hypoxia and/or hypoglycemia) of ischemia alone and in combination. The level of glucose in the brain is lower than the level of plasma glucose (80 mg%), ranging from 20 mg% (for euglycemia) and from 1 to 3 mg% for hypoglycemia [20]. Similarly, the tissue concentration of oxygen corresponds to oxygen levels between 2 and 8% [21], defining multiple layers of conditions beyond an obstructed vessel (Figure 1). In our experiments, beyond normal and hypoglycemic brain conditions, we also modeled normoglycemic plasma glucose conditions (80 mg%). Such higher glucose concentrations are also widely used in *in vitro* neuroscience research.

We chose the concentration of 2 HU/mL, corresponding to a 0.1–0.2 µg/mL amount of the active toxin in the cerebrospinal fluid in patients [6]. Earlier works characterize this amount as “sub-lytic”, indicating acute lysis in <15% of the cells, but with wide-spread non-lytic effects in the rest of the cells [8]. To characterize the gross effect of PLY on cell lysis in tissues at normal peripheral glucose conditions, we analyzed PI permeabilization (a standardized lytic permeabilization assay) in mixed glial cells with 80 mg% glucose (Figure 2A,B). Hypoxia increased lytic pore formation after 120 min (Figure 2A,B). Challenging acute brain slices with 2 HU/mL PLY either in continuous optimal oxygenation (using carbogen (95% O_2_/5% CO_2_) perfusion) conditions for 8 h [22] or on a rotary shaker (in a 5% CO_2_ environment and normal atmospheric oxygen, leading to hypoxia in the slice cultures) for the same time led to a two-fold increase in the PLY-induced LDH release during hypoxia (another lytic permeabilization loss marker), indicative of the presence of elevated delayed (8 h) hypoxic damage (Figure 2C). The challenge with 2 HU/mL PLY, despite the evidence for synaptic loss [12], did not elevate the LDH release when the slices were adequately oxygenized (Figure 2C).

Brain tissue damage in conditions of brain ischemia is glutamate-dependent in the penumbral area around the ischemic core [23]. Additionally, glutamate-dependent neurotoxicity plays a role in the synaptic loss induced by PLY [12]. We exposed hypoxic brain slices to 2 HU/mL PLY either alone or in the presence of 10 µM MK-801 [24]—a selective non-competitive inhibitor of the NMDA receptor, capable of reverting NMDA-mediated excitotoxicity—to test whether the increased hypoxic cytotoxicity caused by PLY at 80 mg% glucose may be due to the excessive glutamate release by the toxin, as observed in earlier works during synaptic damage in meningitis. In hypoxic conditions and during 8 h of incubation, MK-801 did not block PLY-mediated cytolysis in the hypoxic slices, and even enhanced it (Figure 2C), eliminating the key role of glutamate as a factor, and promoting acute cell damage in ischemia after the PLY challenge.

Next, we switched our cell model system to glucose levels, corresponding to real brain tissue normo/hypoglycemic conditions [25]. We incubated mixed glial cultures with a combination of several conditions—namely normoxia/normoglycemia (8% O_2_/20 mg% glucose), hypoxia/normoglycemia (2% O_2_/20 mg% glucose), normoxia/hypoglycemia (8% O_2_/3 mg% glucose), and hypoxia/hypoglycemia (2% O_2_/3 mg% glucose). Direct irreversible permeabilization of the glial cells induced by PLY was analyzed via PI staining [8]. After the challenge with 2 HU/mL PLY, reducing the glucose to hypoglycemic levels enhanced lytic permeabilization two-fold (Figure 3). Approximately 80% of the lysed cells demonstrated a microglial morphology. Reducing the oxygen level to hypoxic conditions, however, did not enhance permeabilization, indicating that it was not oxygen but rather the glucose level that was more critical (Figure 3). The reduction in oxygen from normoxic to hypoxic conditions delayed the permeabilization half-time, indicating delayed toxin binding and/or pore formation (Figure 3).

Next, we analyzed the effects of hypoxia/ischemia on subtler non-lytic changes in the cells after the PLY challenge—by analyzing the calcium (Ca) influx. Using a precise approach to discriminate astrocytes and microglia established before [26], we analyzed Ca increases using the Cal520 sensor dye following the 2 HU/mL PLY challenge (Figure 4). In single microglial cells, the cellular calcium increased following PLY exposure to a plateau and either decreased or remained high without strong fluctuations at the plateau (Figure 4A). In individual astrocytes, the pattern differed—due to syncytium-like interconnection, the increase followed a wave (“flickering”) pattern with elevations and drops within short periods (Figure 4B). In individual astrocytes, the magnitude of the peaks was lower and shorter than in the microglia.

Pooled together, the microglia demonstrated, in conditions of normoxia and normoglycemia, a rapid Ca increase, followed by restoration of the Ca levels at 400 s. For all test groups, the cumulative Ca elevation following the PLY challenge was always significant versus the normal non-treated background Ca levels (Figure 4C,D). In all other conditions, peaks were lower and the half-times of the increase were longer, indicating slower/weaker binding of the PLY (Figure 4C). Especially weak was this effect in isolated hypoglycemia, which contrasted with its increased PI permeabilization (Figure 3). In microglia, there was a rapid recovery of Ca after the peak in the normoxia/normoglycemia group. This was much slower in the hypoxia and hypoglycemia group. In the isolated hypoxia and hypoglycemia groups, the elevated Ca did not show any restoration (Figure 4C).

In the astrocytes, PLY initiated a cumulative Ca increase under normoxic/normoglycemic conditions, followed by a recovery, but this was much slower than that in microglia (Figure 4D). The pattern of the Ca increase in the astrocytes comprises multiple oscillating peaks, which overlap; therefore, the cumulative curves show higher variability (see Figure 4B). In all other conditions, recovery of the Ca peak was missing (Figure 4D). As in the microglial Ca analyses, the PLY challenge in isolated hypoglycemia led to milder, but stable, peaks, indicating weaker/slower toxin pore formation, but missing the adaptive normalization response (Figure 4D). Comparing the Ca increases, the levels in microglia were significantly higher than those in astrocytes (Figure 4E). In the Appendix A (8% O_2_/20 mg% glucose) and Appendix A (8% O_2_/20 mg% glucose), the difference in the speed and the peak of calcium influx, and the restoration, being strongly diminished and delayed in hypoglycemia, are visible.

To summarize, in normal conditions, sub-lytic PLY initiated in the glial cells Ca elevation, followed by recovery. PLY in isolated hypoglycemia elevated the permeabilization of the glia, but the non-permeabilized cells demonstrated weaker Ca elevation with no recovery. In all hypoxic conditions (with or without hypoglycemia), PLY led to moderate Ca elevation, but impaired recovery (all summarized in Figure 4F).

To clarify the cause of this stronger elevation in Ca in microglia than in astrocytes, we used filipin-based staining of cholesterol (PLY receptor) (Figure 5). The microglia demonstrated a significantly higher cholesterol content (Figure 5A,B), suggesting a higher toxin-binding capacity. When using recombinant PLY-EGFP, the microglia (Figure 5C,D (outline)) demonstrated indeed much higher toxin binding in many cells (Figure 5C,D (red arrows)).

Next, we analyzed the basic endocytosis rate in glial cells, using a fluorescent endocytotic FM4-64 assay. Membrane turnover acts as a repair system for pore-damaged membranes [27]. Hypoglycemia reduced the endocytotic turnover by half (Figure 6).

To clarify the role of hypoxia and/or hypoglycemia in cellular traffic phenomena as a factor in the observed toxin effects, we used human red blood cells, known to be largely devoid of endocytotic membrane turnover (making them also much more hemolytically vulnerable against PLY). Here, ischemic effects should be mediated predominantly by direct toxin/membrane interaction modulation. While in normoxic conditions, no change in the lysis was observed between different glucose conditions (Figure 7), during hypoxia, diminished lysis was observed at low glucose (3 mg%) concentrations, indicating directly inhibited PLY/membrane interactions (Figure 7).

## 3. Discussion

Our work showed that changes in tissue glucose and oxygen levels modulate PLY’s lytic and non-lytic pore-forming properties. While hypoxia enhanced pore formation in normal peripheral glycemic conditions, in hypoglycemic brain conditions, it even reduced it. Generally, with a reduction in glucose levels, the toxin pore-forming effects diminished, as well as the adaptive responses (such as endocytosis) of the brain tissue. Throughout our experiments, we analyzed two major patterns of cell damage—direct acute lysis (in normal conditions in <15% of the cells) and non-lytic Ca elevation in most of the other cells.

Brain ischemia, vascular inflammation, and secondary brain infarction during meningitis were described more than a century ago [28,29,30]. In experimental animal models of pneumococcal meningitis in rats, cortical necrosis in places of brain infarction is used as a disease progress readout [31]. Subcortical white-matter ischemia and secondary axonal damage discretely accompany pneumococcal meningitis too [13]. The overexpression of HIF-1α as a sign of hypoxia is widely observed in meningitis samples, even in areas without necrosis, indicating wider-spread penumbral conditions [15]. Complex brain perfusion changes, not only due to inflammation but also due to the increased intracranial pressure in the closed rigid skull, show that maintaining normal pressure and better perfusion correlates with better survival [32,33]. While in areas of necrosis due to ischemia the tissue is irreversibly damaged, the penumbral area can be still rescued and the role of factors influencing its fitness is very important.

PLY is well known as a critical toxic factor of pneumococcus, and when it is missing, the course of disease is much milder, independent of the organ [3,34,35]. PLY is a pore-forming toxin, but at disease-relevant concentrations (0.1–0.2 µg/mL [6]), it leads to minimal lysis and affects cellular functions without lysing cells [12]. PLY alone causes brain swelling [36] and exacerbates inflammation too, as all of these effects remain pore-dependent. Cell lysis represents an irreversible loss of vitality without the ability to restore cellular function. Non-lytic pores, in contrast, can be repaired and cells can further function, although multiple cellular functions are altered [37]. PLY causes long-term alteration to small-GTPase activation and cellular trafficking changes too [10,38]. Therefore, it is critical to know how all pathogenic factors alter the transition from sub-lytic to lytic toxin performance. Several elements influence acute PLY lytic effects—such as the amount of toxin, the number of binding receptors (cholesterol) on the cell surface, and the ability of cells to compensate transient pore formation and repair membranes via endo/exocytosis and membrane vesicle shedding [37,39]. Within several hours following non-lytic pore formation, secondary cell damage because of Ca overload may occur, switching reversible cellular alterations into irreversible ones [40,41]. The exact mechanisms involving Ca in these processes apart from “classical” apoptosis include regulated necrosis (such as necroptosis, pyroptosis, and ferroptosis) too [42]. The ability of host cells to restore elevated cytosolic Ca levels (e.g., following PLY exposure), therefore, plays a role in the prevention of long-term damage [43].

We outlined several major dependencies of PLY activity on glucose and oxygen levels, consistent with normal and ischemic conditions in non-neural tissues and in the brain. Hypoxia exacerbated the lytic damage caused by PLY in normal peripheral glucose conditions, but in hypoglycemia, this effect vanished, and hypoxia even turned paradoxically into being protective. The major receptor for PLY and other members of the cholesterol-dependent cytolysin group of toxins is cholesterol [44]. Therefore, most logically, a diminished/slower effect during hypoxia should be sought in the altered membrane cholesterol distribution. Hypoxia leads to the internalization of membrane cholesterol in lung cells [45]. Several works suggest the role of cholesterol-rich caveolin-containing microdomains (rafts) in oxygen level sensing, but the exact mechanism remains unclear [46]. Half of all membrane cholesterol is sequestered into plasma membrane sphingomyelin rafts and the role of hypoxia can be mediated through its redistribution too [47]. In hypoxia, however, sphingomyelinase activation occurs, and cholesterol is rapidly liberated into the biologically active pool [48], which does not agree with our findings of decreased PLY binding during hypoxia and hypoglycemia. One reason for this is that we modeled both components in our cultures—oxygen and glucose supplementation, independently. In the brain, however, they are dependent on one another (due to autoregulation), as a reduction in oxygen levels leads to a spontaneous reduction in glucose in the extracellular space too [25]. To dissect the role of membrane lipid endocytotic traffic alterations, we switched to human erythrocytes, which do not readily endocytose and possess limited capacity to alter toxin membrane receptors via cell trafficking changes. The effect of glucose reduction was only mild during hypoxia, indicating that a substantial portion of our findings include diminished cell adaptive responses. Still, hypoxia and hypoglycemia together were capable of diminishing toxin/membrane interactions.

In conditions of increasing cell glucose and oxygen deprivation, Ca-level restoration (a sign of cellular fitness) after its toxin-induced influx slowed continuously. Ca-level restoration requires intact ATP/metabolic status, and a lack of oxygen plus lower glucose levels apparently impaired these homeostatic mechanisms [49]. This remained, however, not so apparent due to a concomitant reduction in PLY’s effects caused by hypoxia (overview in Figure 8). The reduction in the endocytosis rate during hypoxia is also relevant to the capacity of cells to repair damage—in conditions blocking endocytosis, the effect of pore-forming toxins is enhanced, suggesting one more mechanism modulating PLY’s effects [39].

Within the glial cell population, microglia demonstrated a higher cholesterol content, a higher permeabilization rate, a stronger Ca influx, and stronger PLY binding than astrocytes. Monocytes/macrophages are known to be selectively targeted by CDCs [50] and our results showed a higher PLY-binding ability of the microglia—a cell type with a similar origin to that of monocytes/macrophages [51]. Indeed, not all cellular functions require strong toxin binding—astrocytes, even when binding less toxin, exhibit strong cell-shape changes in response to it [36]. Still, in the experiments with filipin-stained cholesterol, we observed two microglial cell populations with apparently different cholesterol contents. This differentiation of the two populations was also visible in the binding of fluorescent toxin, which also demonstrated microglial cells with lower and higher toxin binding. It can be speculated that permeabilization increased in cells with higher toxin binding during hypoglycemia due to diminished repair, while in cells with lower toxin binding, hypoxia diminished the binding even further, resulting in a slower Ca influx. At the same time, the restoration of elevated Ca, especially in the microglia, was also delayed or completely missing during hypoxia. Thus, even when the effect of PLY during hypoxia was initially weaker, the total Ca balance was disturbed and the Ca level was elevated (Figure 8).

The experiments with acute brain slices recapitulated the general gross effect of PLY in tissue ischemia, demonstrating increased damage in normal glucose conditions. While in intact animal/human brain conditions, glucose is lower than in cultures and in peripheral tissues [20], our acute brain slice system was established with higher glucose levels comparable to peripheral tissues. Despite this, it was useful as (i) a tissue model system; (ii) a translational system, allowing comparison between brain and peripheral glucose concentration conditions. Earlier work indicates that synaptic loss and damage during meningitis and in acute brain slice systems challenged with PLY is glutamate NMDA receptor-dependent [12]; therefore, we had to test whether these tissue effects were NMDA-dependent too. Indeed, this was not the case, indicating that the exacerbation of PLY-induced tissue damage by hypoxia under normoglycemic conditions is most probably relevant to other non-neural tissues too. This is not surprising, as the pore-forming effects of PLY are glutamate-independent. Still, additional experiments are needed to examine the sensitivity and relevance to various peripheral tissues.

Methodologically, our work outlines the possibility for certain differences between toxin effects under 80 mg% and 20 mg% glucose levels. While in native brains, 20 mg% is the normal glucose level, in vitro, >80 mg% is the established culture condition. In cases such as hypoxia and PLY exposure, this discrepancy can produce contradictory results and needs to be considered when interpreting experimental outcomes between cell culture and animal/human systems.

Our experimental results suggest that vascular, circulatory, and metabolic properties of the host and tissue microenvironment alter PLY’s effects. This introduces an additional heterogeneity of effects during infections with *S. pneumoniae* and needs to be considered. In other diseases, where CDCs play a pathogenic role, this can be of importance for the development of symptoms (if these toxins demonstrate similar dependance). In gas gangrene, for example, where *C. perfringens* and its CDC perfringolysin play major roles, oxygen deprivation and ischemia are critical for disease progression. Animal model studies and analyses of other members of the CDC group are needed to address the pathogenic significance of this dependance.

## 4. Materials and Methods

### 4.1. Recombinant PLY Preparation

Wild-type PLY and PLY-EGFP were expressed in E. coli BL-21 cells (Stratagene, Cambridge, UK) and purified using metal affinity chromatography as described in detail previously [52]. The purified PLY was tested for Gram-negative LPS contamination using a colorimetric LAL assay (KQCL-BioWhittaker, Lonza, Basel, Switzerland). All purified proteins showed <0.6 endotoxin units/µg of protein. Hemolytic activity was evaluated based on a standard assay described previously [53]. Briefly, one hemolytic unit (HU) was defined as the minimum amount of toxin needed to lyse 90% of 1% human erythrocytes per ml within 1 h at 37 °C. In hemolysis experiments, absorption at 492 nm was measured. Equivalent lytic capacity in red blood cells does not explicitly correspond to equivalent lytic capacity in other cell types. For various batches of PLY, we determined hemolytic capacity between 40,000 and 100,000 HU/mg. In all experiments, the concentration of the toxin was presented as HU/mL.

### 4.2. Cell Culture and Tissue Culture

Primary mouse astrocytes and microglia were prepared as mixed glial cultures from the brains of newborn C57Bl/6 mice (postnatal day (PD) 3–5) through mechanical dissociation and 2 rounds of washing in PBS, followed by incubation in Dulbecco’s modified Eagle medium (high in glutamate) (Gibco, Thermo Fisher Scientific, Waltham, MA, USA), supplemented with 10% heat-inactivated fetal calf serum (FCS) (PAN Biotech GmbH, Aidenbach, Germany) and 1% penicillin/streptomycin (Gibco). Cells were seeded in poly-L-lysine (PLO, Sigma-Aldrich Chemie GmbH, Taufkirchen, Germany)-coated 75 cm^2^ cell culture flasks (Sarstedt AG & Co KG, Nuembrecht, Germany) until day 14, when the astrocytes formed a confluent monolayer with clearly distinguishable microglia on the top of the layer. Cells were harvested by trypsinization and subsequently cultured in 4- or 8-well chamber slides (Sarstedt) for further experiments. Culture treatment with PLY was performed in serum-free medium.

Acute brain slices were prepared from infant (postnatal day 10–14) C57Bl/6 mice or Wistar rats via decapitation and vibratome sectioning (Vibroslice NVSL, World Precision Instruments, Berlin, Germany) in artificial CSF continuously oxygenized with carbogen gas (95% O_2_, 5% CO_2_) at 4 °C. The slices were then allowed to adapt in carbogenated Basal Medium Eagle (Gibco) with 1% penicillin/streptavidin and 80 mg% glucose at 37 °C for 1 h before being incubated in pure BME supplemented with 80 mg% glucose. In some experiments, the brain slices were oxygenated with carbogen continuously during exposure to the toxin or mock [22]. For experiments in hypoxic conditions, slices were incubated in the same medium on a rotary shaker (60 rpm/min) in a 5% CO_2_ incubator, with each slice being individually incubated in a separate well of a 24-well plate (Sarstedt).

### 4.3. LDH Assay

Lactate dehydrogenase (LDH) release as a sign of brain tissue lysis was analyzed according to the manufacturer instructions of an LDH assay (Promega AG, Dübendorf, Switzerland). Completely lysed brain slices were used as a reference for 100% lysis, following the recommendation of the company.

### 4.4. Cell Culture Ischemia Model System

Brain ischemia is accompanied not only by hypoxia of the affected tissue but also by hypoglycemia [18]. To maintain the glucose level in the cultures as close as possible to that of native brain tissue conditions, we used DMEM supplemented with 20 mg% glucose (Sigma) for euglycemia and 3 mg% for hypoglycemia. For precise establishment of the tissue gas-level conditions, we used a mix of 8% O_2_, 5% CO_2_, and 87% N_2_ for normoxic conditions (at normal atmospheric conditions, the cultures were hyperoxic) and 2% O_2_, 5% CO_2_, and 93% N_2_ for hypoxia modeling (premixed gasses, Westfalia AG, Kirchberg, Switzerland). We did not use an anoxia model due to the lack of rescue of such tissues (if present) without specific treatment, which is outside the scope of this work.

Cells were incubated in a custom-built microscopy chamber, using a gas-impermeable plastic folio covering the top of a PeCon heating plate (PeCon GmbH, Erbach, Germany). The bottom was tightly closed with a 0.17 borosilicate coverslip to allow imaging without changing the atmosphere. The chamber was continuously flushed with the corresponding gas mix with a pressure of 1 atmosphere, using a gas humidifier (Carl Roth GmbH + Co. KG, Karlsruhe, Germany) and incubating all cells for at least 2 h before starting the experiments. All treatments were performed using insulin syringes with an insulin needle (B. Braun Medical AG, Sempach, Schweiz), punching directly through the cover folio and using a standardized volume of 100 µL/treatment.

### 4.5. Microscopy Live Imaging Assays (Permeabilization, Calcium Measurement, Endocytosis)

Imaging was performed on an Olympus Cell^M imaging system (Olympus Deutschland GmbH, Hamburg, Germany) using 10× and 20× objectives with continuous maintenance of the temperature at 37 °C and while incubating the cells in DMEM without phenol red, supplemented with glucose as described above. We used fluorescence intensities of 7 and 11% and a camera exposure time of 30–50 ms, with filters for DAPI, DsRed, and EGFP, corresponding to the fluorescence of the fluorophores used. Permeabilization was measured using propidium iodide (PI) and general nuclear stain Hoechst333 (Life Technologies, Thermo Fisher Scientific, Waltham, MA, USA) according to an established procedure and a final concentration of 1 µg/mL, following PI staining at 4 min intervals.

Calcium imaging was performed using Cal520 AM stain (AAT Bioquest, Pleasanton, CA, USA), preloaded for 60 min in DMEM, and then washed and incubated again in fresh DMEM. Images were taken using an EGFP filter every 10 s for 150 s before the challenge with 1 HU/mL PLY and 150 s after the challenge. Intensity was measured after background correction and normalization to the average intensity of the cells before toxin exposure. Astrocytes, forming a confluent monolayer, and microglia, which have a different morphology and demonstrate a higher resting calcium concentration, were discriminated according to an established protocol [26].

For analysis of the background endocytosis rate, we used an FM4-64 assay [10]. Following 2 h of preincubation with normal or low-glucose DMEM, 5 µg/mL FM4-64 (N-(3-triethylammonium propyl)-4-(6-(4-(diethylamino)-phenyl)-hexatrienyl) pyridinium dibromide) fluorescent dye was added. Endocytosis was measured at a single time point at 2 h, as the fluorescence followed continuous elevation as shown in Appendix A after addition of the stain.

### 4.6. Filipin Staining

Mixed glial cultures in 8-well chamber slides were fixed with 3% formalin for 1 h at room temperature. Following extensive rinsing, the cells were incubated with 0.1 mg/mL filipin (Sigma) in PBS for 2 h at room temperature in the dark. Following extensive rinsing, samples were imaged under the Olympus fluorescent microscope with a 340/380 nm filter set.

### 4.7. Statistical Analyses

Statistical analyses were performed using GraphPad Prism 9.3.0 for Windows (GraphPad Software Inc., La Jolla, CA, USA). Statistical comparison of two groups was performed using parametric (for all analyses with *n* > 10) Student *t*-tests or paired *t*-tests (if performed in parallel); non-parametric (for all analyses with *n* < 10) Mann–Whitney U-tests; or paired Wilcoxon tests (if performed in parallel, which most of the live imaging experiments were). These analyses were performed comparing every pair of groups differing in one parameter (e.g., toxin, oxygen, or glucose) instead of a one-way ANOVA for multiple group comparisons, since an ANOVA requires only one variable factor when comparing multiple groups (e.g., different concentrations of the same substance). For non-linear regression, one-phase exponential association demonstrated the highest match of fitting of the permeabilization curves. For the analysis of the calcium influx, we used bell-shaped fitting. All values represent the mean ± SEM. All tests were performed as two-tailed tests. *p*-values below 0.05 were considered statistically significant using the following *p*-value classification—* *p* < 0.05, ** *p* < 0.01, *** *p* < 0.001, **** *p* < 0.0001.

## Figures and Tables

**Figure 1 toxins-16-00232-f001:**
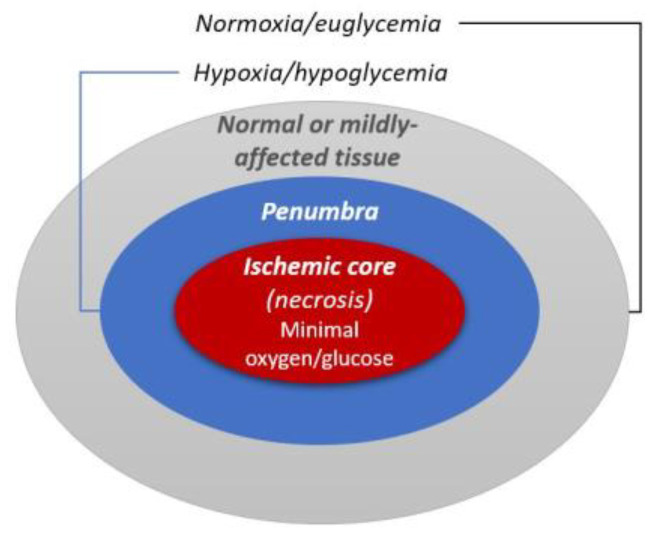
Schematic representation of different tissue layers surrounding an ischemic core.

**Figure 2 toxins-16-00232-f002:**
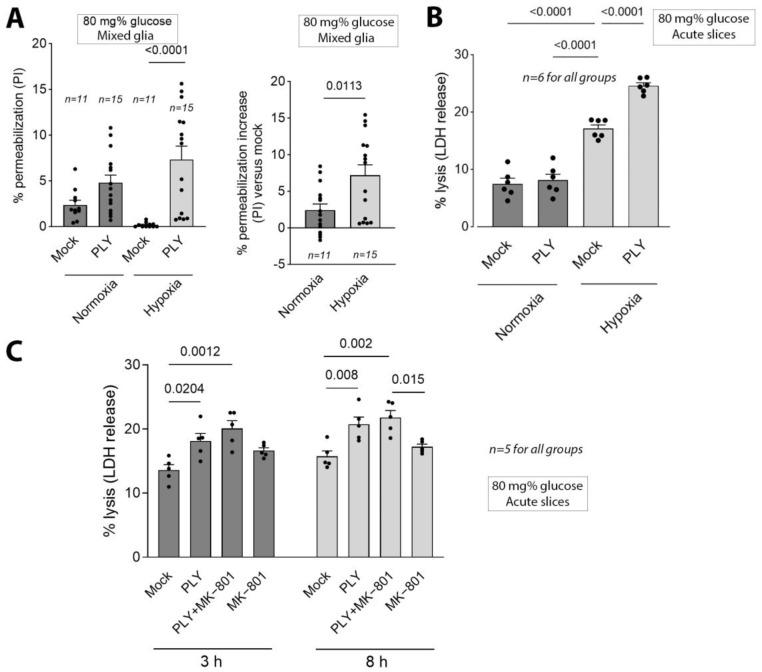
Lytic effects in acute brain slices in normoxic and hypoxic environments. (**A**) Propidium iodide (PI) permeabilization of mixed glia following 2 HU/mL PLY in normoxic and hypoxic conditions, showing increased permeabilization during hypoxia. Glucose level at 80 mg%, comparable with a normal peripheral (non-brain) tissue glucose concentration. (**B**) Exposure to 2 HU/mL PLY in acute slices with active perfusion with carbogen (95%O_2_/5% CO_2_) (normoxia) and with reduced oxygen levels (on a shaker in a 5% CO_2_ incubator with normal atmospheric air). Lysis was measured 8 h following each PLY challenge according to the release of LDH (lactate dehydrogenase) in the medium. (**C**) Effect of 10 µM MK-801 on the lytic properties of 2 HU/mL PLY in the acute slices incubated in hypoxic conditions. The elevated lysis in the presence of PLY was not reverted. Brain slices were incubated at a glucose level of 80 mg%. All values represent the mean ± SEM; groups are compared using a one-way ANOVA; values above the bars indicate *p* when significant; *n* indicates independent experiments.

**Figure 3 toxins-16-00232-f003:**
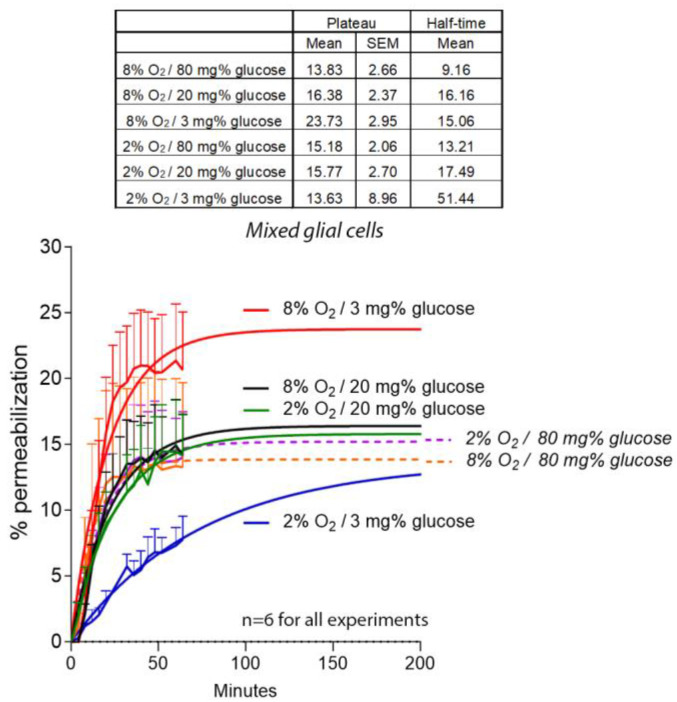
Lytic effects of PLY in different conditions of hypoxia/hypoglycemia in mixed glia. Dynamic propidium iodide-based permeabilization of mixed primary mouse glial cells following exposure to 2 HU/mL PLY. Only in conditions of isolated hypoglycemia was permeabilization elevated. In dual hypoxic/hypoglycemic conditions, permeabilization was diminished and the half-time was elevated, indicative of slower toxin binding and/or pore formation. For all groups, *n* = 6 independent experiments.

**Figure 4 toxins-16-00232-f004:**
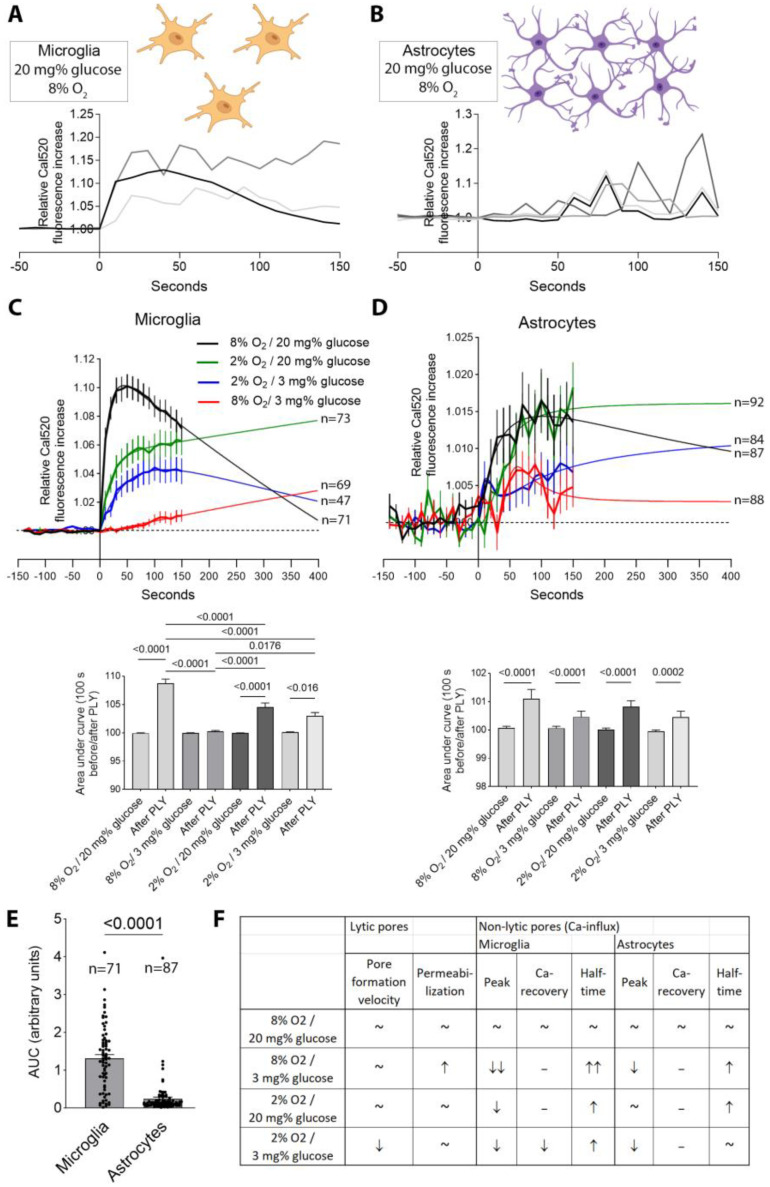
Calcium elevation in mixed glia caused by PLY. (**A**) Cal520 fluorescence increase as a marker of Ca elevation in 3 individual microglial cells from primary glial cultures (2 HU/mL PLY was applied at timepoint 0) for demonstrative purposes. Schematic cell images from BioRender. (**B**) Cal520 fluorescence increase following PLY exposure in 4 astrocyte cells. Note the difference in the pattern of Ca fluctuation due to the syncytium-like interconnections between the astrocytes, allowing very flexible Ca buffering and wave distribution. (**C**) Cumulative curves of Ca elevation in microglia following exposure to 2 HU/mL PLY; *n* indicates the number of cells, pooled together from 4 independent experiments. Below the curves, a statistical comparison of the areas under the curves is presented. (**D**) Cumulative curves of Ca elevation in astrocytes following exposure to 2 HU/mL PLY; *n* indicates the number of cells, pooled together from 4 independent experiments. Below the curves, a statistical comparison of the areas under the curves is presented. (**E**) Comparison of the area under the curve (AUC) between the cumulative elevation of Ca induced by PLY in microglia and in astrocytes in conditions of normal oxygen and normal glucose, indicating a stronger effect in microglia. All values represent the mean ± SEM; groups are compared using the Mann–Whitney U-test; values above the bars indicate *p* when significant; *n* indicates individual cells; pooled together from 3 experiments. (**F**) Tabular presentation of the Ca changes in microglia and in astrocytes under different conditions ((~) indicates normoxic/normoglycemic conditions and their equivalents).

**Figure 5 toxins-16-00232-f005:**
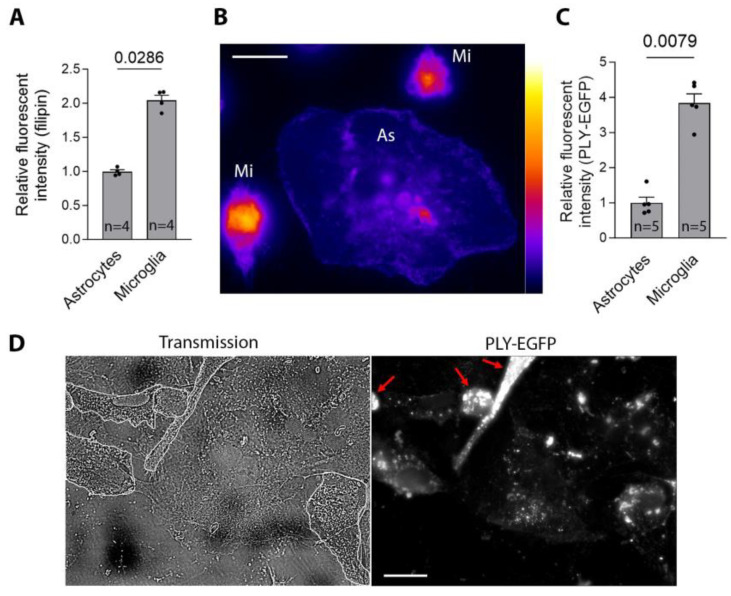
Cholesterol content in microglia and astrocytes. (**A**) Relative fluorescent intensity of filipin staining as a marker of cholesterol content in microglia and astrocytes, demonstrating a higher content in the microglia. (**B**) False-color fluorescent filipin image of an astrocyte (large cell, As) and microglia (Mi). (**C**) Relative fluorescent intensity of the bound PLY-EGFP after 10 min of exposure (4 HU/mL), demonstrating increased PLY binding in the microglia. (**D**) Images of microglia/astrocyte co-culture with the microglia outlined (left, transmission image). Microglial cells with exceptionally strong PLY-EGFP binding are indicated with arrows (fluorescent image of PLY-EGFP, right). All values represent the mean ± SEM; groups are compared using the Mann–Whitney U-test; values above the bars indicate *p* when significant; *n* indicates independent experiments. All scale bars: 20 µm.

**Figure 6 toxins-16-00232-f006:**
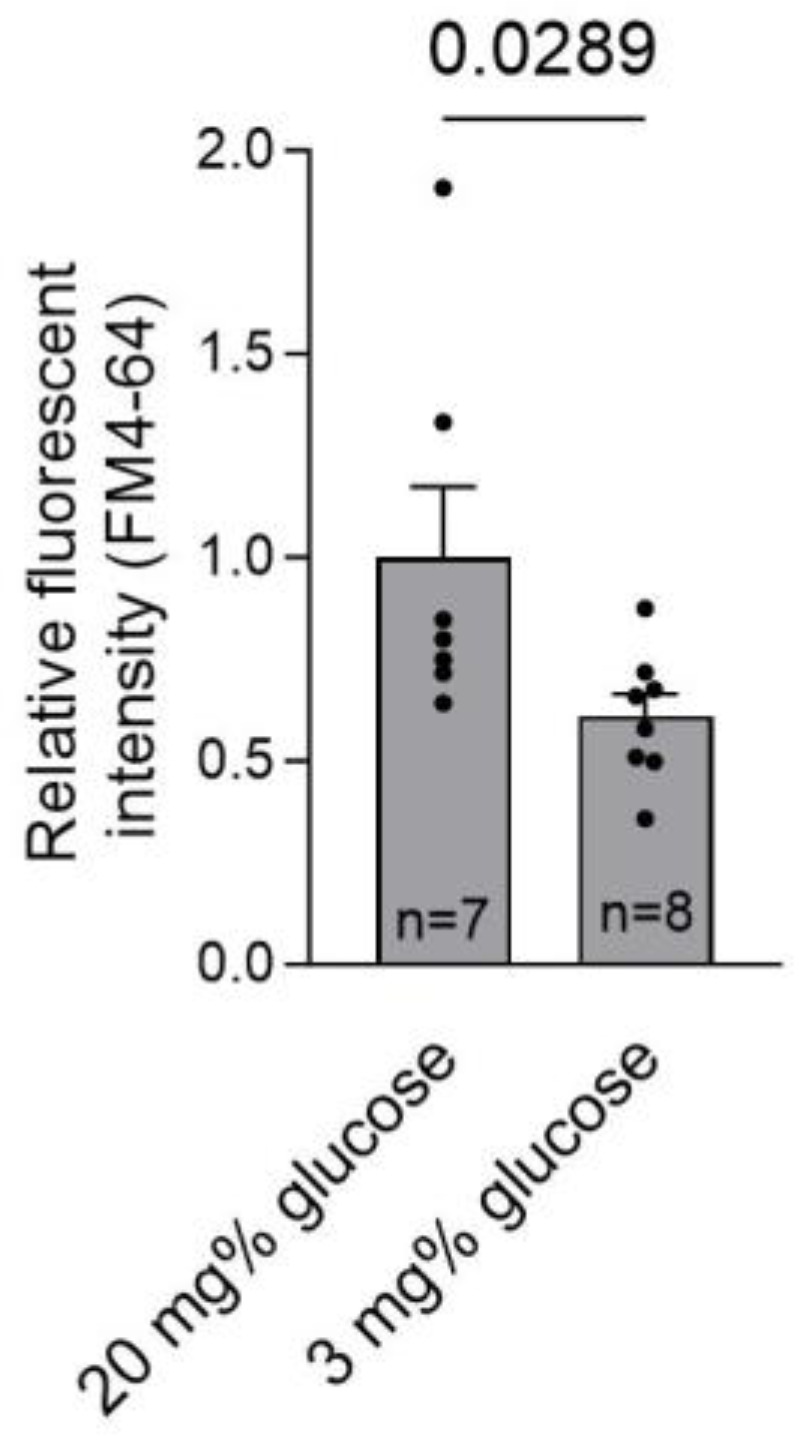
Diminished endocytosis rate during hypoglycemia. Following 2 h of cell conditioning at 20 mg% (normoglycemia) and 3 mg% (hypoglycemia), the basic FM 4-64 turnover was recorded and compared between normoglycemic brain conditions (20 mg%) and hypoglycemic brain conditions (3 mg%). All values represent the mean ± SEM; groups are compared using the Mann–Whitney U-test; values above the bars indicate *p* when significant; *n* indicates independent experiments.

**Figure 7 toxins-16-00232-f007:**
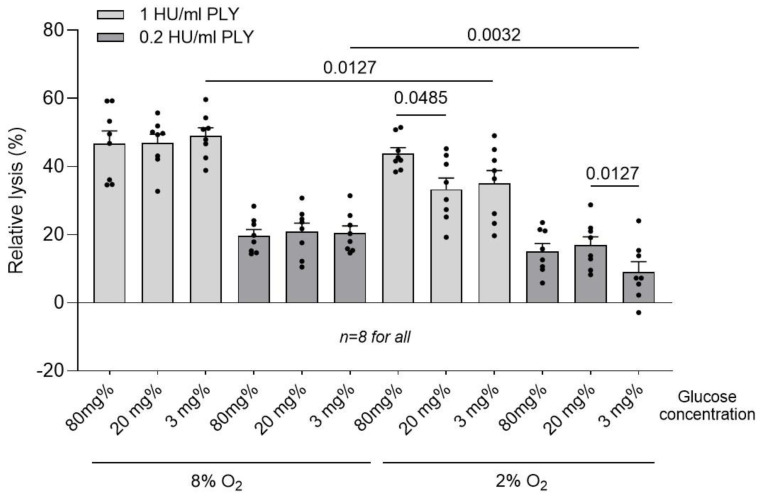
Effect of 0.1 and 1 HU/mL PLY on red blood cell hemolysis under various glucose conditions, and in normoxic and hypoxic environments (37 °C, 30 min). While in normoxia, glucose variation did not produce any change in hemolysis, in hypoxia, diminishing glucose concentrations demonstrated a mild reduction in the toxin’s effects. All values represent the mean ± SEM; groups are compared using the Wilcoxon paired test; values above the bars indicate *p* when significant; *n* indicates independent replicates.

**Figure 8 toxins-16-00232-f008:**
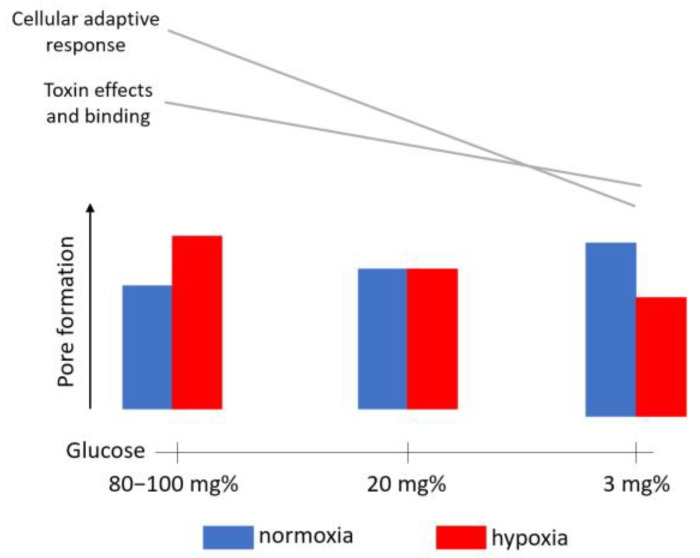
Schematic presentation of the major changes from normoxia/normoglycemia to hypoxia/hypoglycemia (ischemia).

## Data Availability

The generated datasets can be found at https://doi.org/10.48620/404.

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
