# Peer review of "Glucose and Oxygen Levels Modulate the Pore-Forming Effects of Cholesterol-Dependent Cytolysin Pneumolysin from Streptococcus pneumoniae"

_toxins, 2024, doi:10.3390/toxins16060232_

Round 1
Reviewer 1 Report
Comments and Suggestions for Authors
Glucose and oxygen levels modulate pore forming effects of the 2 cholesterol-dependent cytolysin pneumolysin from Streptococcus pneumoniae is an interesting manuscript investigating the contribution of glucose and oxygen availability to S. pneumoniae’s primary toxin, pneumolysin’s activity on brain cells, specifically astrocytes and microglia. The authors evaluated permeabilization of cells under varying conditions of normal and hypoglycemia as well as normoxic and hypoxic conditions. Additionally, the authors evaluated cholesterol content of cells as a potential reason for differences in toxin-mediate permeability between cell types. The findings provide new information regarding potential effects of PLY on host cells. However, there are several concerns regarding the experiments and presentation of the data.
Major points:
1) The authors use 2HU/mL PLY throughout the study. However, the first studies (Figures 2 and 3 are aimed at looking at effects on cell lysis. Then in Figure 4 they state they are investigating “subtler-non lytic changes in cells (calcium influx). However, they are using the same dose of PLY (2HU/mL). If this dose is capable of causing cell lysis, how is the same dose subtler? If the length of time is the determining factor, this should be clarified. The authors do not provide PI or LDH results for the early time points investigated in Figure 4. Perhaps the cells are also lysed at this early time point. This should be addressed.
2) The authors state that Figure 2A demonstrates that hypoxia increased lytic pore formation. However this seems to be dependent upon the decrease in permeabilization in the Mock cells under hypoxic conditions. The authors don’t directly indicate a significant difference between the PLY-treated samples in normoxic and hypoxic conditions, though there is an increase. Also, in what should be Figure 2C, the Mock hypoxic cells had more LDH release than the normoxic Mock cells which doesn’t match with the PI staining in Figure 2A. Also, the increase in LDH release due to PLY could simply be PLY potentiating whatever is leading to this hypoxic Mock cell LDH release.
3) Data in Figure 4A is of only 3 individual microglial cells. Yet later in the discussion of the fluorescent staining they state that there was considerable variation in 2 populations of microglial cells with higher and lower PLY-binding. How with only an n=3 cells do they know they are seeing representative results.
Minor points:
1) Supplemental movies are not discussed in the results and should be omitted if not contributing to the interpretation of the results.
2) Line 39 says PLY produces cell host lysis. Perhaps this should be host cell lysis.
3) The results describe Figure 2D, but there are only A, B, and C panels. It appears that 2C is really 2D and 2B is 2C.
4) NMDA-receptor is not well-described. The authors use an inhibitor of NDMA-receptor but do not clearly state why. The association of this receptor with glutamate needs to be fleshed out more as well as their hypothesis for using the inhibitor in the study.
5) What should be Figure 2 D only shows 3 and 6hr data but the results say the experiment was done for 8Hrs.
6) In Line 135-136, authors state that cumulative Ca-elevation following PLY challenge was always significant (not shown). Why is this significance not indicated or shown in supplemental data?
7) The methods state that the data in Figure 6 was done at multiple time points, but only 2hr data is shown. Fluorescence measurements at other time points should at least be included in supplemental data if not Figure 6.
8) Some description of the results seems contradictory. For example, in the abstract the authors state: “continuously diminishing toxin-pore forming capacity in cells when reducing oxygen. Then in Line 197-198 of Discussion they state that hypoxia enhanced pore formation in normal glycemic conditions.
Author Response
Glucose and oxygen levels modulate pore forming effects of the 2 cholesterol-dependent cytolysin pneumolysin from Streptococcus pneumoniae is an interesting manuscript investigating the contribution of glucose and oxygen availability to S. pneumoniae’s primary toxin, pneumolysin’s activity on brain cells, specifically astrocytes and microglia. The authors evaluated permeabilization of cells under varying conditions of normal and hypoglycemia as well as normoxic and hypoxic conditions. Additionally, the authors evaluated cholesterol content of cells as a potential reason for differences in toxin-mediate permeability between cell types. The findings provide new information regarding potential effects of PLY on host cells. However, there are several concerns regarding the experiments and presentation of the data.
Major points:
1) The authors use 2 HU/mL PLY throughout the study. However, the first studies (Figures 2 and 3 are aimed at looking at effects on cell lysis. Then in Figure 4 they state they are investigating “subtler-non lytic changes in cells (calcium influx). However, they are using the same dose of PLY (2HU/mL). If this dose is capable of causing cell lysis, how is the same dose subtler? If the length of time is the determining factor, this should be clarified. The authors do not provide PI or LDH results for the early time points investigated in Figure 4. Perhaps the cells are also lysed at this early time point. This should be addressed.
Answer: The concentration of 2 HU/ml corresponds to 0.1-0.2 µg/ml at the activity of the recombinant toxin we use, and it has been established as comparable to clinically relevant PLY concentrations in meningitis before (Spreer et al., 2003). This is the reason for choosing it. In series of earlier works, the effect of this concentration on permeabilization has been extensively tested to characterize lytic/sub-lytic effects (Foertsch et al., 2011; Iliev et al. 2008). Shortly, in these works comparison between permeabilization (as judged by PI staining), LDH, fluorescent stains was performed. The data evidenced that we have two distinct populations - permeabilized (PI-positive and fluorescent stain-negative) cells (defined within 15%), and a larger portion of non-permeabilized live cells that remain intact, but respond with transient pore formation and massive changes in the normal function - diminished motility, reorganization of actin and tubulin cytoskeleton, increased phagocytosis (references included in the text). The permeabilization with PLY in our experiments follow exactly the same curve of sub-lytic damage as the works cited above, indicating that in all conditions (hypoxia and/or hypoglycemia) the permeabilization behavior is similar. In this manner, we are able to mimic best the small lysed and the larger altered, but not lysed populations, observed in real tissue. We add this information more precisely as references.
2) The authors state that Figure 2A demonstrates that hypoxia increased lytic pore formation. However this seems to be dependent upon the decrease in permeabilization in the Mock cells under hypoxic conditions. The authors don’t directly indicate a significant difference between the PLY-treated samples in normoxic and hypoxic conditions, though there is an increase. Also, in what should be Figure 2C, the Mock hypoxic cells had more LDH release than the normoxic Mock cells which doesn’t match with the PI staining in Figure 2A. Also, the increase in LDH release due to PLY could simply be PLY potentiating whatever is leading to this hypoxic Mock cell LDH release.
Answer: Indeed, there are variations between different conditions (hypoxic and normoxic) regarding the mocks, which, we think, has several reasons - in Fig. 2A, we use dissociated glial cultures, in Fig. 2B and C - acute brain slices. In acute slices, there is always a certain level of stronger damage (mock at 10% versus <1% for dissociated cultures), therefore comparing mock conditions between slices and dissociated cultures is not straight-forward. Indeed, we have noted the seemingly paradoxical finding of lower PI-staining in the mock hypoxia mixed glia group versus the normoxic mock group, which seem consistent through the cultures. Still, these variations are within the low percentages (0-1% for hypoxic mock, and 0.5-3% for normoxic mock), with only 2 values beyond, the rest still within the SEM. It seems, however, that hypoxia shortly and paradoxically boosts phagocytosis of damaged cells by microglia, which has been described for macrophages before (PMID: 17675562). Still, it falls beyond the scope of our work and we do not discuss it in detail. For us, the key in these experiments is the comparison with real-time identical conditions controls. In the case of the dynamic curves (Fig. 3), we subtracted the mock control PI values for each group, eliminating possible confusion about less PI-stained cells in hypoxic mock.
3) Data in Figure 4A is of only 3 individual microglial cells. Yet later in the discussion of the fluorescent staining they state that there was considerable variation in 2 populations of microglial cells with higher and lower PLY-binding. How with only an n=3 cells do they know they are seeing representative results.
Answer: Here, the three cells are just an example of the various patterns without drawing major conclusions on them, we just show that the pattern of individual astrocytes and individual microglia differ. In the following diagrams (Fig. 4C/D), we built the cummulative curves based on >60 cells, where we see many more cells, defining these two populations, outlined as single cells in A and B.
Minor points:
1) Supplemental movies are not discussed in the results and should be omitted if not contributing to the interpretation of the results.
Answer: Corrected. In the text, exact description and the rationale behind their inclusion in the submission is outlined.
2) Line 39 says PLY produces cell host lysis. Perhaps this should be host cell lysis.
Answer: Corrected.
3) The results describe Figure 2D, but there are only A, B, and C panels. It appears that 2C is really 2D and 2B is 2C.
Answer: Corrected, 2D refers to an earlier version of the figure.
4) NMDA-receptor is not well-described. The authors use an inhibitor of NMDA-receptor but do not clearly state why. The association of this receptor with glutamate needs to be fleshed out more as well as their hypothesis for using the inhibitor in the study.
Answer: Corrected, we extended the text clarification about it.
5) What should be Figure 2 D only shows 3 and 6hr data but the results say the experiment was done for 8Hrs.
Answer: Corrected, 8 h is the correct time-point.
6) In Line 135-136, authors state that cumulative Ca-elevation following PLY challenge was always significant (not shown). Why is this significance not indicated or shown in supplemental data?
Answer: Statistical comparisons of the areas under the curve is now indicated in the panels of Fig. 4C and D, as well as in the text.
7) The methods state that the data in Figure 6 was done at multiple time points, but only 2hr data is shown. Fluorescence measurements at other time points should at least be included in supplemental data if not Figure 6.
Answer: Here, we made a mistake in the Methods description. We established the FM4-64 staining protocol for measurement of the endocytosis rate in glial cells, which demonstrates that FM4-64 continously accumulates into the cells (now included in Fig. 6). At 2 h, we performed snapshots of several treated cultures to make a comparison between thei basic endocytotic rate. Now, we add the basic FM elevation curve as a supplementary image after adding the stain and following it to demonstrate the dynamics of change.
8) Some description of the results seems contradictory. For example, in the abstract the authors state: “continuously diminishing toxin-pore forming capacity in cells when reducing oxygen. Then in Line 197-198 of Discussion they state that hypoxia enhanced pore formation in normal glycemic conditions.
Answer: Here, the text should indicate that hypoxia enhances pore formation in normal peripheral glycemic conditions. The three modalities of glucose levels - 80, 20 and 3 mg%, are characterized with differences in hypoxia effect. We are aware of the complexity of the dependencies discovered by us, and we now made definitions more consistent throughout the text.
Reviewer 2 Report
Comments and Suggestions for Authors
In the article titled "Glucose and oxygen levels modulate pore forming effects of the cholesterol-dependent cytolysin pneumolysin from Streptococcus pneumonia", the authors aimed at studying the effect of oxygen and glucose levels on the PLY-mediated damage to the brain tissue. The paper is well written and well presented. There are a couple of points that I wold like the authors to address:
1. Clarify the rationale of using 2HU/mL dose of PLY. Was a dose-dependent experiment done to determine the sublytic concentration of PLY?
2. In the discussion, the authors did touch a note of the effect of hypoglycemia and hypoxia on the cholesterol in the host cell membrane. It would be be important and crucial to include another CDC like a SLO to show a similar effect seen with PLY in terms of cell lysis. It seems that the effect is not specific to PLY but would be seen across different CDCs. Alternatively, is there a mutant form of PLY available that is defective in cholesterol-binding?
3. Is there any role of hypoxia or hypoglycemia on the activation of GTPases like Cdc42 or rac which are involved in the internalization of damaged GPIs/endocytosis induced by toxins? That is something that could be discussed in the discussion.
Comments on the Quality of English LanguageSome grammar needs to be corrected. Overall, it reads very well.
Author Response
In the article titled "Glucose and oxygen levels modulate pore forming effects of the cholesterol-dependent cytolysin pneumolysin from Streptococcus pneumonia", the authors aimed at studying the effect of oxygen and glucose levels on the PLY-mediated damage to the brain tissue. The paper is well written and well presented. There are a couple of points that I wold like the authors to address:
- Clarify the rationale of using 2HU/mL dose of PLY. Was a dose-dependent experiment done to determine the sublytic concentration of PLY?
Answer: The exact sublytic concentration was determined in series of previous works, namely - inducing lytic effects below 15% of all cells with non-lytic changes in the rest (Foertsch et al., 2011; Iliev et al. 2008). These non-lytic effects include cytoskeleton changes, calcium increase, cellular stress response, and all of them are translated in disease-relevant effects. 2HU/ml PLY corresponds to 0.1-0.2 µg/ml PLY, but variations from a batch to a batch of recombinant toxin exist, therefore we and others sticked to hemolytic units as more precise and reliable parameter. For our recombinant batches, we observe minimal variations of 20-30%, but from a lab to a lab, such variations can be really high. When compared with toxin concentration in the cerebrospinal fluid of patients with pneumococcal meningitis, concentrations in the range of 0.1 to 0.3 µg/ml were detected, with hemolytic activity similar to our recombinant toxin. Thus, we believe we use amounts of toxin
- In the discussion, the authors did touch a note of the effect of hypoglycemia and hypoxia on the cholesterol in the host cell membrane. It would be be important and crucial to include another CDC like a SLO to show a similar effect seen with PLY in terms of cell lysis. It seems that the effect is not specific to PLY but would be seen across different CDCs. Alternatively, is there a mutant form of PLY available that is defective in cholesterol-binding?
Answer: We would like to thank the reviewer for these questions, which are very intriguing, but may require several additional lines of experiments. To answer some of the most relevant ones, which are mechanistically very important, we performed an alternative experiment. Shortly, we used cells that cannot perform endocytosis and have minimal ability to adapt the membrane cholesterol levels by subcellular membrane trafficking - red blood cells. In them, the effects of PLY should be purely toxin/membrane-dependent and membrane recycling independent - i.e. no cholesterol internalization is possible. These experiments confirmed that hypoxia and glucose together diminish toxin effects, but alterations in the cellular membrane traffic are as important - in erythrocytes, the role of hypoxia and hypoglycemia (only during hypoxia) was milder.
- Is there any role of hypoxia or hypoglycemia on the activation of GTPases like Cdc42 or rac which are involved in the internalization of damaged GPIs/endocytosis induced by toxins? That is something that could be discussed in the discussion.
Answer: This is extended and added to the discussion.
Round 2
Reviewer 1 Report
Comments and Suggestions for Authors
The authors have addressed the concerns.